# Real-Time Human Action Recognition with a Low-Cost RGB Camera and Mobile Robot Platform

**DOI:** 10.3390/s20102886

**Published:** 2020-05-19

**Authors:** Junwoo Lee, Bummo Ahn

**Affiliations:** 1Robotics Group, Korea Institute of Industrial Technology, Ansan 15588, Korea; leejw0@kitech.re.kr; 2Robotics & Virtual Engineering, KITECH Campus, Ansan 15588, Korea

**Keywords:** real-time, human action recognition, mobile robot, embedded board, RGB camera

## Abstract

Human action recognition is an important research area in the field of computer vision that can be applied in surveillance, assisted living, and robotic systems interacting with people. Although various approaches have been widely used, recent studies have mainly focused on deep-learning networks using Kinect camera that can easily generate data on skeleton joints using depth data, and have achieved satisfactory performances. However, their models are deep and complex to achieve a higher recognition score; therefore, they cannot be applied to a mobile robot platform using a Kinect camera. To overcome these limitations, we suggest a method to classify human actions in real-time using a single RGB camera, which can be applied to the mobile robot platform as well. We integrated two open-source libraries, i.e., OpenPose and 3D-baseline, to extract skeleton joints on RGB images, and classified the actions using convolutional neural networks. Finally, we set up the mobile robot platform including an NVIDIA JETSON XAVIER embedded board and tracking algorithm to monitor a person continuously. We achieved an accuracy of 70% on the NTU-RGBD training dataset, and the whole process was performed on an average of 15 frames per second (FPS) on an embedded board system.

## 1. Introduction

As the aging population and one-person households increase worldwide, societal and political aspects affect health care. Owing to these effects, the health care models are changing from in-person care to remote monitoring using smart sensors, as the number of seniors per caregiver are increasing [1]. The key feature of the remote monitoring care is home-based care, in which the activities of people are monitored at home to prevent emergencies. Therefore, several studies have proposed various approaches as per the type of manipulation of the sensors and largely categorized them into vision and nonvision based.

The most common nonvision based sensors such as accelerometers, motion sensors, biosensors, gyroscopes, and pressure sensors are wearable and can be attached to the users like a daily-use object. The most common approach of using multimodal sensors is by placing them in a person’s living environment such as in the kitchen or the living room to record their daily routine activity [2]. The sensors monitor activities by achieved sensor data, such as when a person uses the items, such as opening doors (switch sensor), sitting down on the couch (pressure sensor). Although this is a non-intrusive way of collecting sensor data, these types of sensors have an on-off functionality, which makes it difficult to recognize detailed actions. Torres-Huitzil et al. have presented a smartphone-based action recognition system using an embedded accelerometer sensor, which allows free positioning of the smartphone on the human body [3] and Ahmed et al. presented hybrid feature selection on smartwatch sensor data, which outperforms other state-of-the-art models [4]. In another approach based on IoT technique, they monitored the vital signs remotely using a wearable sensor to classify the activities [5]. Chung et al. built a mobile device and body-worn inertial measurement unit sensors for activity data collection [6]. Machine learning approaches are widely used, automatic labeling method for data annotation is presented [7]. A major drawback of these approaches is that the sensors that they use are attached to the body of the user which makes them uncomfortable, and they are unable to recognize detailed actions due to the performance of the sensors. For this reason, recent studies have used camera sensors for recognizing human action, and these studies have been increasingly dealt with in the field of computer vision and machine learning.

Previous studies started vision-based approaches by designing hand-crafted features (optical flow) such as histograms of oriented gradients (HOG) [8], histograms of optical flow or motion boundary histograms (MBH) [9] that are extracted from 2D images, and the actions are classified by support vector machine. However, the implementation of these hand-crafted features was difficult, and was, therefore, limited in the experimental environment. The deep-learning technique has exhibited superior performances in image recognition, speech recognition, natural language processing, and human action recognition. Thus the improvement of classification strategies using a convolutional network and temporal segment network (TSN) was proposed [10]. TSN achieved enhanced results; however, it is based on optical flow and could not provide a real-time process. To overcome the bottleneck of calculating optical flows, Zhang et al. proposed a method using a motion vector to replace optical flow, enabling knowledge transfer from the optical flow domain to the motion vector domain [11]. The processing time was further improved, but optical flow data from the training data is still needed. Recent research on human action recognition using deep learning was classified into two categories owing to the characteristics of the image data. In the case of image recognition, it was classified via learned kernels by extracting features such as spatial information from a single image. In the case of video data, because it is composed of a sequence of images, the current image was determined using a former image, and the temporal information. Therefore, at first, two convolutional neural networks (CNNs) were used to learn spatial information through RGB images, and the temporal information was learned by accumulating the optical flow and average of two class scores [12]. Using two CNNs, end-to-end learning was infeasible, and the speed of optical flow calculation of the images was insufficient for real-time use. Correspondingly, based on the idea of an attempt to extend 2D convolutional structures to 3D spatiotemporal structures, 3D convolutional networks were developed (C3D) [13]. They found that they have the best temporal kernel length and that the 3-D system outperforms 2D convolutional features on video analysis tasks. They also reported the runtime analysis on a single K40 Tesla GPU processing at 313 FPS. However, according to a paper that characterized several commercial edge devices on different frameworks and well-known convolutional networks, the C3D network is computationally intensive and requires high memory usage, which is not possible with the embedded board system [14]. Secondly, Song et al. considered the recurrent neural network (RNN) that extracts temporal features from the skeleton coordinate, which was obtained using Microsoft’s Kinect camera [15]. The Kinect camera uses the time-of-flight method to obtain depth information for characterizing the skeleton joints once the pulsed infrared light (emitted by an IR projector) is reflected back to the object. Using this method, it can easily detect 6 people at once and 25 joints per person. Because of this effective 3D pose estimation technique, many benchmark datasets have been made using this camera and several studies have been performed based on RNNs because of its strength in dealing with sequential data and skeletal data. The attention module was attached to select and focus on joints and frames by providing a level of importance for each action. In other words, the spatial and temporal information was considered by focusing on joints and frames. However, due to the use of public data made by Kinect camera to obtain 3D skeletal data, the learning of various action labels has been restricted. Besides that, the above studies could not be used for real-time classification, in particular, on the embedded system, because of the complexity of the model structure that is required to achieve a high recognition score for given benchmark datasets. Moreover, a neural network trained on a Kinect dataset must be used by a Kinect camera and the size of this camera is not feasible for loading on a mobile robot.

In this study, we proposed a method that can recognize human actions in real-time with a mobile robot on an embedded system. Considering the size of the mobile robot, we used a single RGB camera instead of the Kinect camera to achieve 3D skeleton joints by integrating the two open-source libraries. In this way, the range of use of the training dataset increases by using not only the public Kinect dataset but also the dataset that achieved by RGB camera like previously recorded video. To use a simple image classifier of the CNN model for an embedded system, we converted the skeleton joint data to image form. The whole process is depicted in Figure 1. Skeletal joints were extracted from the RGB camera image, and their movements were classified using CNN within the embedded board. For continuous observation during indoor conditions, we can track the person using the skeleton joints.

The rest of this paper is organized as follows. In Section 2, we have described how to achieve a 3D skeleton joint on a single RGB camera, and have presented a CNN structure with the conversion of skeleton joint data into image form, and a mobile robot platform with a tracking algorithm. In Section 3, the experimental setup is described and in Section 4, the results and discussion of the trained NTU-RGBD dataset and tracking have been analyzed. Finally, conclusions have been presented in the last section.

## 2. Materials and Methods

In this section, at first, we have introduced the method of extracting a 3D skeleton joint using a single RGB camera instead of a Kinect camera, and later, we have described the converting process of skeleton data to image form for using it in the embedded system. Afterward, a simple CNN model architecture has been introduced, and finally, we have described the mobile robot platform and tracking algorithm.

### 2.1. Pose Estimation Pipeline

Recently, there have been various studies for human pose estimation, which is key for the understanding of human behavior on a robotic system. 2D pose estimation has made significant progress via OpenPose [16]. Today, many researchers are devising methods for estimating 3D postures with a single camera. Several applications have been performed; however, there are few methods for real-time performances. Habermann et al. first showed a real-time full-body performance capture system using a single camera [17]. They proposed an innovative two-stage analysis that reconstructs the dense, space-time coherent deforming geometry of people in loose clothing. For the pipelined implementation, they used two NVIDIA GTX 1080Ti GPUs [18] and achieved around 25 FPS. Dushyant Mehta et al. presented a more improved method with a real-time approach for multi-person 3D motion capture at over 30 FPS using a single RGB camera [19]. They contributed a new CNN network called SelecSLS net to improve the information flow. For real-time performances, they used a single NVIDIA GTX 1080Ti GPU. All these approaches show great real-time performances; however, the hardware setup does not apply to the embedded system.

To run on the embedded system, 3D skeleton joints were extracted on RGB camera without depth information, by integrating two open-source libraries such as OpenPose and 3D-baseline [20]. Both libraries are based on deep-learning, and OpenPose is a state-of-the-art architecture of pose estimation. It can detect multi-person without limits and can estimate 2D key points of the human body, such as hand and face. 3D-baseline is a lifting algorithm that predicts a 3D position on a given 2D joint location. We have integrated two libraries into one pipeline so that the RGB images from the camera are passed through the OpenPose to obtain a 2D joint, and this joint input to the 3D-baseline can be lifted to 3D coordinate. To do this, OpenPose and 3D-baseline were trained on the CoCo dataset [21] and MPII dataset [22], respectively, and we had to set the joint ordering equally.

We have adjusted the 2D joint output of OpenPose to the 3D-baseline input format. Therefore, as shown in Figure 2, we obtained a total of 17 joints in 3D coordinate using a single RGB camera without a depth sensor.

### 2.2. Skeleton to Image Conversion

Recent trends of human action recognition research are based on the RNN model which uses skeletal data as input data; however, as we have mentioned previously, their model complexity increases thus making end-to-end learning difficult. Among the area of deep-learning networks, image recognition fields such as GoogleNet [23] and Resnet [24], state-of-the-art models based on the CNN model, have shown great performances as it became easier to train on their own dataset as an open-source library. For this reason, many studies are converting pose sequence data to images to use CNN [25].

To use these CNN models as a classifier of action recognition, we converted the 3D joint input data coordinates x, y, z into red, green, blue pixel values of an RGB image, respectively, to form a single image. The shape of the joint from a video clip consisting of one action is an M × N × 3 matrix form of Equation (1), where M is the number of frames and N is the number of joints. In this study, we have used body parts such as hip, right hip, right knee, right foot, left hip, left knee, left foot, spine, thorax, neck/nose, head, left shoulder, left elbow, left wrist, right shoulder, right elbow, and right wrist, in the order of the output of 3D-baseline.
(1)$$S=\left(\begin{array}{ccc} x_{1}{}^{1}y_{1}{}^{1}z_{1}{}^{1} & \cdots & x_{1}{}^{N}y_{1}{}^{N}z_{1}{}^{N}\\ \vdots & \ddots & \vdots \\ x_{M}{}^{1}y_{M}{}^{1}z_{M}{}^{1} & \cdots & x_{M}{}^{N}y_{M}{}^{N}z_{M}{}^{N} \end{array}\right)$$

To convert these matrix values into RGB values, we extracted the values of *x*, *y*, *z* from matrix S, and normalized them in the continuous range from 0 to 255. After normalization, the converted RGB matrix from a single frame was 1 × *N* × 3, and by stacking the frames by column, we obtained a single image size of *M* × *N* with RGB channels as shown in Figure 3. In this study, we used a sliding window to generate a subsequence in a video clip that contains one action for data augmentation. We set the stride to 1 for the sliding window to get the predicted action label on each frame, and the window size to 30 frames to execute approximately one to two seconds of movement in an image. Consequently, the action label on the current frame is a result of the previous 30 frames. A converted image of a standing action sequence is depicted in Figure 4. During the action, we can locate the change of color in the columns.

### 2.3. CNN Model

The classification of actions has been made easier by using 6 convolutional layers for using it in a real-time embedded system. As shown in Figure 5, it has a 3 × 3 kernel with padding 1 and stride 1. We have attached batch normalization [26] after the second and fourth layers for the distribution of outputs. In addition, after the second and fourth layers, a 2 × 2 max pool, and after the last convolutional layer a 2 × 2 global average pool was placed to reduce the size. To initialize weights for all convolutional layers, we have used Xavier initialization [27] and ReLu [28] as an activation function. In the final layer, *k* action labels were classified by a fully connected layer. To prevent overfitting, we also conducted dropouts after the first, third, and fully connected layers with a 70% probability.

### 2.4. Mobile Robot Platform

For the purpose of continuous detection and monitoring of a person’s action in the in-doors, we have placed a tracking algorithm in the mobile robot. The mobile robot used in the experiment consists of NVIDIA JETSON XAVIER board [29], OpenCR [30], ROBOTIS Turtlebot [31], and Logitech c920, as shown in Figure 6. Xavier board is an embedded Linux computer, which is the latest addition of the Jetson platform that enables robotic platforms in the field with workstation-level performance. It comprises a 512-core Volta GPU including 64 Tensor cores, 16 GB memory, and an 8-core ARM v 8.2 64-bit CPU. We performed the whole process of joint extraction and action recognition using this board at an average of 15 FPS To control Dynamixel motors [32], we used serial communication with OpenCR and Xavier board, and for tracking, we used the x coordinate of the center of mass of all the joints extracted from OpenPose and the y coordinate of the height of the detected person. These values were estimated from the embedded board and communicated to OpenCR on a serial circuit. The motors were controlled using the tracking algorithm to keep the detected person in the center of the camera screen and maintain the distance between the person and the robot.

## 3. Experimental Setup

### 3.1. Dataset

The proposed method was evaluated on the NTU-RGBD dataset [33], which is the largest dataset collected by the Kinect V2 camera. It contains approximately 56,000 action sequences from 60 action classes. The most challenging part of this dataset is that it is typically recorded on a variant view that mainly covers three different views (−45°, 0°, 45°). The dataset has four modalities, depth maps, RGB frames, and IR sequences. In our experiments, we have used the RGB frame modality to extract 3D joints by pipeline and trained them on the CNN network. To generate an image, joints from the OpenPose needed to be fully observed. Thus, we sorted the action classes so that no self-occlusion occurs. The action classes were sorted into 15 labels, such as eating a meal/snack, brushing teeth, drop, sitting down, standing up, hopping, jumping up, making a phone call, playing with the phone, checking the time, rubbing two hands together, saluting, putting the palms together, crossing hands in front, and touching chest. As a result, we used a total of 2217 clips that were further converted to a total of 95,266 images by the sliding window method and we conducted 5-fold cross-validation.

### 3.2. Training

To validate our method, we trained our network for 500 epochs using a stochastic gradient descent [34] optimizer by setting the learning rate to 0.001 and the batch size to 64. To avoid overfitting, we used an L2 regularization [35] with a weight decay of 0.0025. To validate the use of a single RGB camera in the action recognition problem through the learning, we compared two types of joint data that were extracted by the proposed method and by using a Kinect camera. As described earlier, we sorted action labels in which identical self-occlusion did not occur. Moreover, using the Kinect camera output of 25 joints, we set the number of joints to 17 just as with the proposed method. All training processes were run on NVIDIA RTX 2080Ti (11 GB) [36], AMD Ryzen 7 2700 [37], and RAM 24 GB in WINDOW OS with a Tensorflow framework.

## 4. Results and Discussion

In this section, we present the training result of the proposed model on the NTU-RGBD dataset with 3 types of data RGB camera-17, Kinect-17, and Kinect-25. We also show the tracking results and full system that recognizing the actions while tracking a person with a mobile robot.

### 4.1. Results of Proposed Model

The results of the NTU-RGBD trained dataset are shown in Table 1. Original data from Kinect camera with a total of 25 joints showed the best accuracy of 75%. Because it had more joint coordinates than our method, more detailed joint information were assumed to improve accuracy. When the number of joints matched, Kinect data and RGB camera achieved an accuracy of 74% and 71%, respectively. From this result, we have confirmed that the performance of the RGB camera did not drop significantly than the Kinect camera. The confusion matrix with respect to the data obtained from the Kinect camera-17, and RGB camera, respectively, are shown in Figure 7 and Figure 8. In the figure, the vertical axis represents the true label of the NTU-RGB data, and the horizontal axis represents the labels predicted by the model. From the Kinect data, more distinct actions like a salute, and crossing hands in front showed an accuracy of 99% and 90%, respectively. While other similar actions using hands like rubbing two hands together, playing with the phone, and checking time showed an accuracy relatively less than 55%. In addition, the results of the proposed method showed high accuracy for distinct actions such as standing, sitting, hopping, and jumping. However, other actions, in particular, that mainly used the arm joints, showed less accuracy than the Kinect data. We presumed that the accuracy gap occurred because of the pose estimation performance of the OpenPose which was slightly lower than the Kinect camera. Further comparison with other state-of-the-art CNN models are shown in Table 2. The inference performance is different from the Jetson Xavier module power consumption. We measured the FPS on the MAX and 15W power modes. The VGG-19 network [38] has a fully connected layer with a size of 4,096, which consumes a large amount of computational memory, making it hard to classify the actions on the Xavier board. Other CNN networks have a skip connection for fast information flow; however, they still have a latency greater than that of our proposed network.

### 4.2. Results of Tracking

We completed our experiments in indoor conditions by using a mobile robot. Therefore, from the upper left of Figure 9, we can confirm the tracking of the recognized person through the x coordinate value of the center of mass, and the y coordinate value of height. In the first row, as a detected person moves to the left, the center of mass point drops to 221, and the motor drives to the left from the center of the detected person. Similarly, in the second row, as the person moves to the right, motor drives to the right as well. In addition, when the person gets closer to the robot, the motor drives backward to maintain a constant distance. The results of the sample cases of the predicted action label are shown in Figure 10. The first row shows the overall picture of the user performing the action in front of the mobile robot. As the user performs actions, skeleton joints are extracted and the actions are classified by the embedded board. We can confirm the predicted action labels on the top left of the picture on the second row of Figure 10, and 3D skeleton joint on the last row.

## 5. Conclusions

In this study, we proposed a method that recognizes human actions using an RGB camera instead of a Kinect camera that can be applied to a mobile robot platform. We extracted the 3D skeleton joint coordinate using the integrated pipeline of OpenPose and 3D-baseline, and converted the joint data into image form for classifying actions using the CNN model. For evaluation, we trained on the NTU-RGBD dataset with two types of data which were obtained using both the Kinect camera and RGB camera. Therefore, our proposed method using the RGB camera achieved an accuracy of 71%, and Kinect-17 and Kinect-25 achieved accuracies of 74% and 75%, respectively. In our opinion, more joint information and precise joint estimation correlate with accuracy. All the processes of joint extraction and action recognition were conducted on an average of 15 FPS on the Xavier board and was applied to a mobile robot platform to use as a monitoring system in the in-doors. Finally, we presented a human tracking algorithm. With this method, we suggest that more variable action labels can be used by utilizing RGB images via formerly recorded videos that were performed in a real-time embedded system. For future work, optimized networks are needed for higher performances, so that they can be used in real-time within the embedded board. In addition, the main limitation of this method that recognizing the front view of the person, a more precise pose estimation algorithm with robustness has to be developed for self-occlusion problems. In the present study, recognizing the action and tracking is performed on a single person; however, we intend to extend this to recognizing multi-person actions and tracking as well.

## Figures and Tables

**Figure 1 sensors-20-02886-f001:**
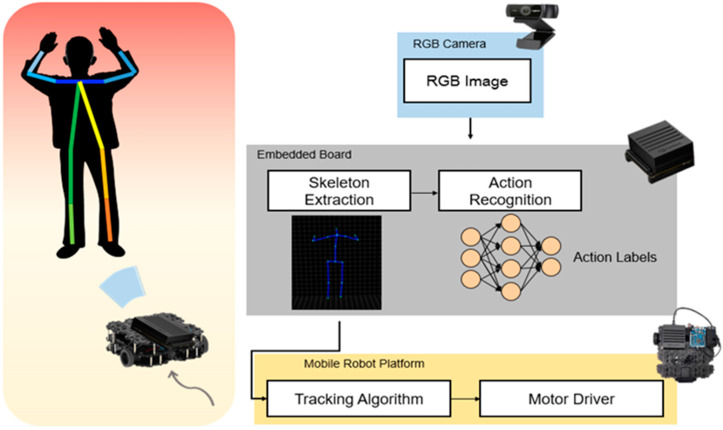
System overview.

**Figure 2 sensors-20-02886-f002:**
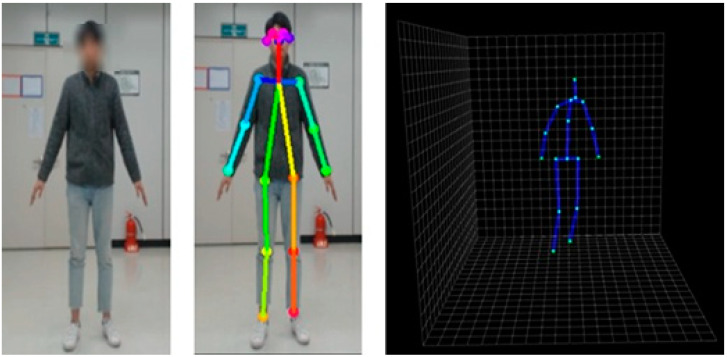
3D skeleton joints obtained by the integrated pipeline.

**Figure 3 sensors-20-02886-f003:**
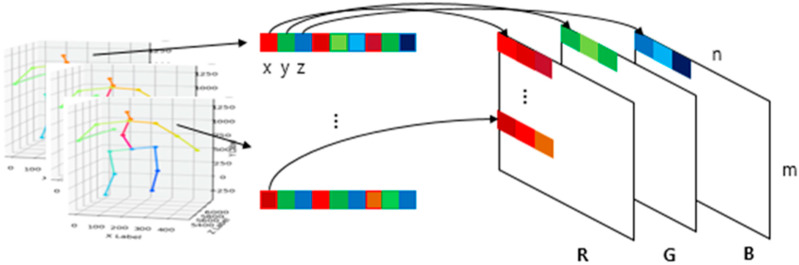
Joint data conversion to RGB image.

**Figure 4 sensors-20-02886-f004:**
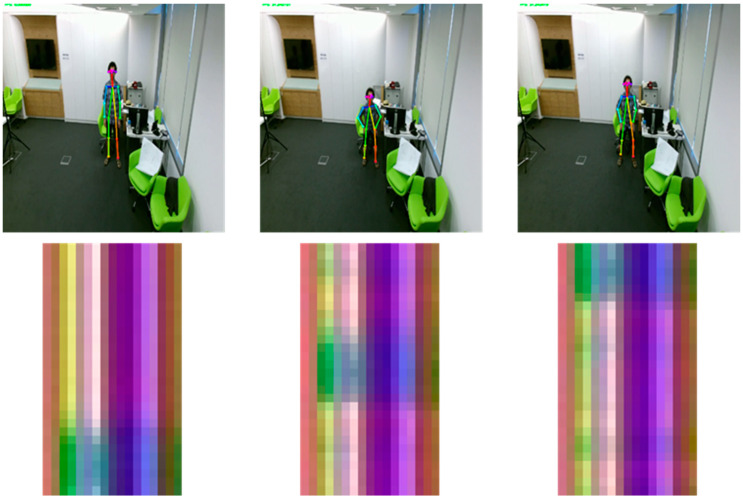
Converted image of the action sequence.

**Figure 5 sensors-20-02886-f005:**
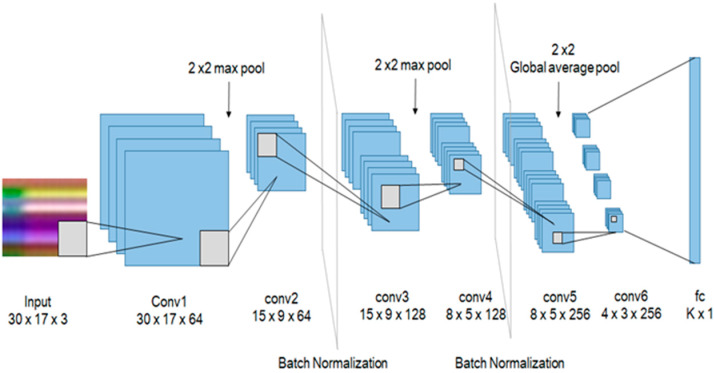
The proposed convolutional neural network (CNN) structure.

**Figure 6 sensors-20-02886-f006:**
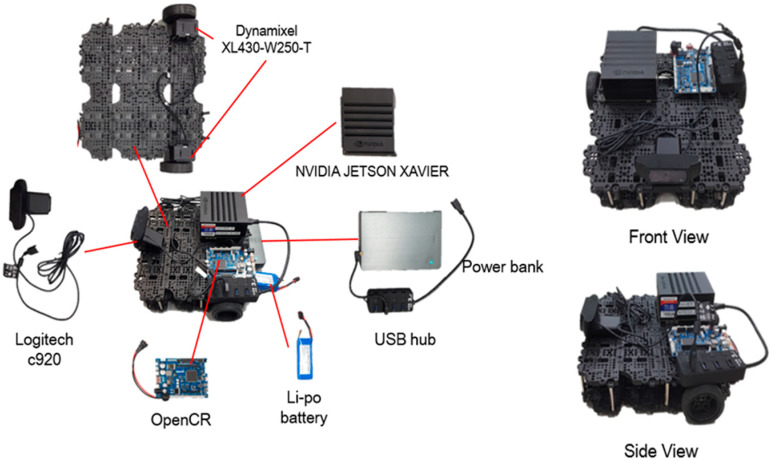
Mobile robot.

**Figure 7 sensors-20-02886-f007:**
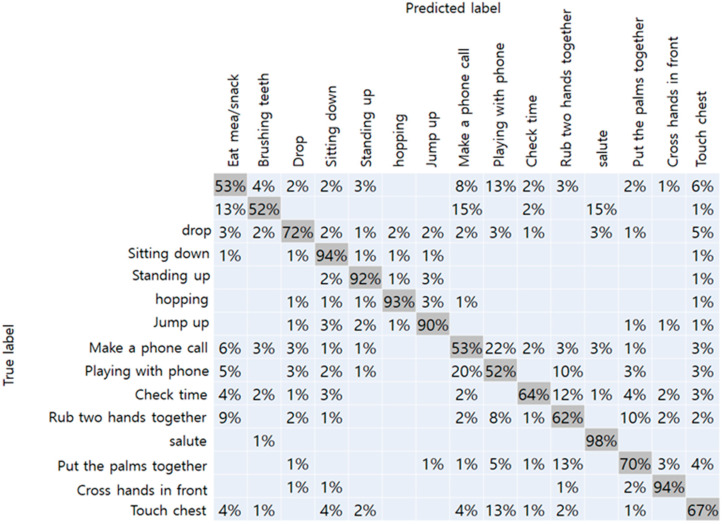
Confusion matrix of Kinect camera-17.

**Figure 8 sensors-20-02886-f008:**
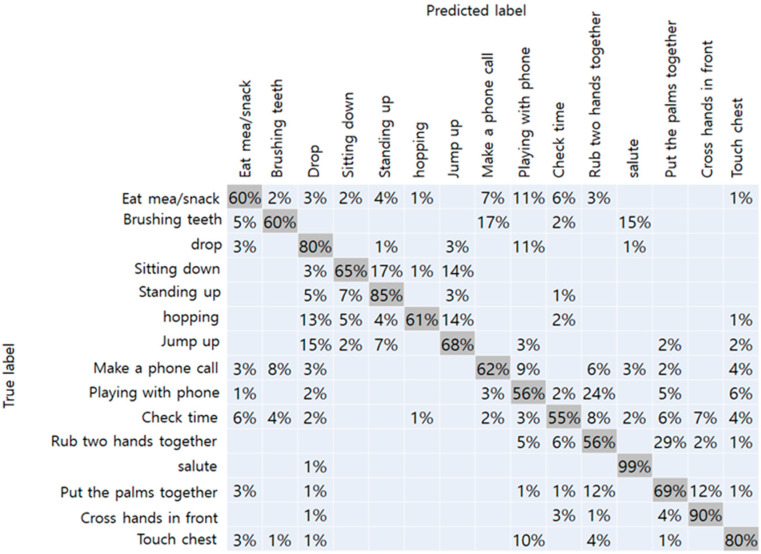
Confusion matrix of RGB camera-17.

**Figure 9 sensors-20-02886-f009:**
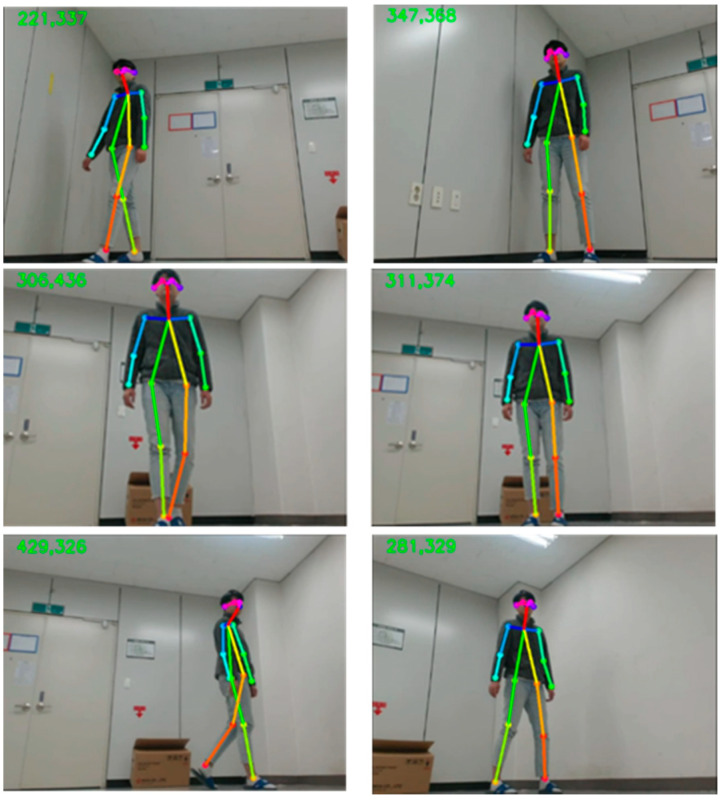
Tracking results during the movement.

**Figure 10 sensors-20-02886-f010:**
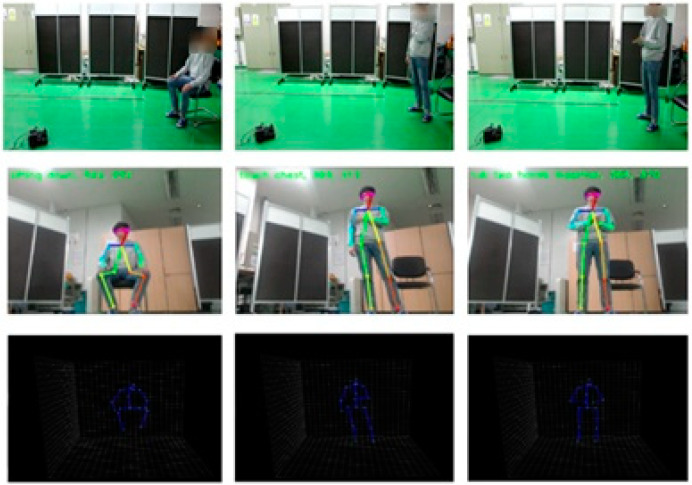
Predicting actions observed by the mobile robot.

**Table 1 sensors-20-02886-t001:** Accuracy by camera and number of joints.

Method	Accuracy	Precision	Recall
**RGB camera-17**	**71%**	**0.71**	**0.69**
Kinect-25	75%	0.74	0.74
Kinect-17	74%	0.73	0.73

**Table 2 sensors-20-02886-t002:** Frames per second between models and module power modes.

Model	15 W Power FPS	MAX Power FPS
VGG 19 [31]	x	x
Inception V4 [22]	4–5	8–9
Resnet-50 [23]	5–6	9–10
**Ours**	**7–8**	**14–15**

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
