# Peer review of "Real-Time Human Action Recognition with a Low-Cost RGB Camera and Mobile Robot Platform"

_sensors, 2020, doi:10.3390/s20102886_

Round 1

Reviewer 1 Report

This paper proposes a novel method of recognizing human actions by using RGB camera, also presents a tracking algorithm. The paper is logical and practical. However, there are some problems to be further improved as well. The comments are:

  1. There are many human action recognition state-of-art literatures based on sensor data. It is better to explain the difference between and the insight why such method proposed by this paper is more beneficial for the Identification. Please explain its advantages in application or other aspects. Some state-of-art literatures in recent 3 years are as following, but not limited to.

- Multi-sensor fusion in body sensor networks: State-of-the-art and research challenges. DOI:10.1016/j.inffus.2016.09.005.

-ShakeIn: Secure user authentication of smartphones with single-handed shakes, IEEE Trans. Mobile Computing, DOI:10.1109/TMC.2017.2651820.

- Imaging and Fusing Time Series for Wearable Sensors based Human Activity Recognition. DOI:10.1016/j.inffus.2019.06.014.

  1. In the experimental configuration, the distribution of training dataset and testing dataset are not clear. Is the evaluation are carried out in cross validation or leave one out? Please explain.

  1. The evaluation criteria is single. Beyond accuracy, recall should also be used to evaluate the performance.

  1. For the validity and applicability of the proposed method, the accuracy of the proposed method with different number of joints can be explored to further demonstrate the performance of proposed model. Otherwise, please explain the reason of joints selection.

  1. There are some typos in the manuscript, such as, in page 8, “as person move to the right” would be “as person moves to the right. Please check the manuscript carefully.

Author Response

Reply to the Reviewers' comments

The authors are grateful for the reviewers’ constructive comments regarding the first manuscript. We have done our best to answer all the questions raised by the reviewers, and have revised the manuscript per their suggestions. We have quoted below all the reviewers’ comments in order, and have included our responses (red) and revised manuscript in the paper (blue) after each comment.

Comments of Reviewer 1

[Comment 1]

There are many human action recognition state-of-art literatures based on sensor data. It is better to explain the difference between and the insight why such method proposed by this paper is more beneficial for the Identification. Please explain its advantages in application or other aspects. Some state-of-art literatures in recent 3 years are as following, but not limited to.

Response) Thank you for your constructive comments. As described in the Introduction section, the approach of human action recognition methods can be classified by the types of sensors used, such as wearable sensors, accelerometers, gyro sensors (with smart phones), and camera sensors. Each approach has its own advantages and disadvantages; however, the major drawbacks occur when sensors are applied in the real world because wearable sensors have to be worn on a person and smartphone-based sensors are largely dependent on the attached location to detect activities. Therefore, recent studies have used camera sensors; however, they mainly focus on the accuracy of a given benchmark dataset, which cannot be applied to embedded board systems. Therefore, in this study, we propose a method that can be applied to an embedded board system with a single RGB camera.

[Comment 2]

In the experimental configuration, the distribution of training dataset and testing dataset are not clear. Is the evaluation are carried out in cross validation or leave one out? Please explain.

Response) Thanks for your comments. We added a detailed evaluation process. The revised manuscripts are in Section 3.1 Line 210.

As a result, we used a total of 2,217 clips that were further converted to a total of 95,266 images with the sliding window method and we conducted 5-fold cross validation.

[Comment 3]

The evaluation criteria is single. Beyond accuracy, recall should also be used to evaluate the performance.

Response) Thank you for your comment. We have added precision and recall in Table 1.

Table 1. Accuracy by camera and number of joints.

Method

Accuracy

Precision

Recall

RGB camera-17

Kinect-25

71%

75%

0.71

0.74

0.69

0.74

Kinect-17

74%

0.73

0.73

[Comment 4]

For the validity and applicability of the proposed method, the accuracy of the proposed method with different number of joints can be explored to further demonstrate the performance of proposed model. Otherwise, please explain the reason of joints selection.

Response) Thank you for your comment. The main reason for joints selection is that our method uses a 3D-baseline toolbox, which outputs 17 joints. The number of joints affects the action recognition outcome. Therefore, we have compared the accuracy between pure Kinect data, which outputs 25 joints and reduced Kinect-17 data in which the numbers of joints are matched with the proposed method. The results are shown in Table 1. Kinect-25 data showed a better accuracy of 75%. Kinect-17 and our method achieved 74% and 71% accuracy, respectively. From this result, we have confirmed that the performance of the RGB camera did not drop significantly more than that of the Kinect camera.

[Comment 5]

There are some typos in the manuscript, such as, in page 8, “as person move to the right” would be “as person moves to the right. Please check the manuscript carefully.

Response) Thank you for your kind attention. We have revised the typos.

Reviewer 2 Report

This paper describes a Human Activity Recognition system for a mobile robot platform using open-source libraries to extract skeleton information and then using CNN for classification. The paper does not provide any new algorithm or system. The main contribution is focused on developing a recognition system of working in real time in a mobile robot. From my point of view, this is a small contribution given the fact that the novelty in the proposal is just using a simpler CNN and a RGB camera.

My main comments to improve the paper are the followings:

  • It is important to remark what is the main contribution of the paper. In the abstract and the end of section 1, the authors explain that they want to develop a system to work in real time. But, during the paper there is not any efficiency analysis or comparison between several strategies.
  • I do not understand very well section 2.2. From RGB images you obtain the skeleton. And then the 3D coordinates are estimated. Then the join points (x, y and z) are translated into RGB images. I do not understand this part and the images showed in fig 3. Are they the results of converting the skeletons? Why do you represent only the skeleton to better understand the relationship? Perhaps, for me the problem is understand why this conversion is necessary?
  • Section 2.3. Have you used dropout for reducing overfitting? Could you specify the tools used for developing the CNN? Are you using keras?
  • Section 3.2. must start with an introductory paragraph.
  • Section 3.2.1. Have you use the same CNN in all cases? The comparison in this section is focused on the camera? The CNN is the same. Have you tried with other CNNs? I think it is important to analyze the time consuming of the different strategies in order to support your contribution.
  • You must include results obtained in previous works using NTU-RGBD dataset.
  • Figure 7 and 8 sizes must be increased. 0% can be removed for a better understanding.

Author Response

Reply to the Reviewers' comments

The authors are grateful for the reviewers’ constructive comments regarding the first manuscript. We have done our best to answer all the questions raised by the reviewers, and have revised the manuscript per their suggestions. We have quoted below all the reviewers’ comments in order, and have included our responses (red) and revised manuscript in the paper (blue) after each comment.

Comments of Reviewer 2

[Comment 1]

It is important to remark what is the main contribution of the paper. In the abstract and the end of section 1, the authors explain that they want to develop a system to work in real time. But, during the paper there is not any efficiency analysis or comparison between several strategies.

Response) Thank you for your constructive comment. The main contribution of this paper is to classify the action in real-time with a mobile robot platform on the embedded board system. We added results from other CNN networks to compare the inference times on the embedded board. The results are shown in Table 2. Section 4.1 Line 252.

Table 2. Frame per second between models and module power modes.

Model

15 W power

FPS

MAX power

FPS

VGG 19 [31]

x

x

Inception V4 [22]

Resnet-50 [23]

Ours

4–5

5–6

7–8

8–9

9–10

14–15

The VGG-19 network [31] has a fully connected layer size of 4096, which consumes a large amount of computational memory, making it hard to classify the actions on the Xavier board. Other CNN networks have a skip connection for fast information flow; however, they still have greater latency than that of our proposed network.

Also, we added previous studies of state-of-the-art action recognition systems that are using HPC (High Performance Computing) to process in real-time, but they are not possible with the embedded board. We have added the manuscript in Section 1 Line 73

Correspondingly, based on the idea of an attempt to extend the 2D convolutional structures to 3D spatiotemporal structures, 3D convolutional networks were developed (C3D) [13]. They found that they have the best temporal kernel length and that the 3D system outperforms the 2D convolutional features on video analysis tasks. They also reported the runtime analysis on a single K40 Tesla GPU processing at 313 FPS.

[Comment 2]

I do not understand very well section 2.2. From RGB images you obtain the skeleton. And then the 3D coordinates are estimated. Then the join points (x, y and z) are translated into RGB images. I do not understand this part and the images showed in fig 3. Are they the results of converting the skeletons? Why do you represent only the skeleton to better understand the relationship? Perhaps, for me the problem is understand why this conversion is necessary?

Response) Thank you for your constructive comment. Figure 3 is the images of converting the 3D skeletons. To classify the actions on CNN networks, joint data to image conversion was needed. There are some advantages of using CNN rather than RNN networks. First, state-of-the-art CNN networks show great performance in the image recognition field and are easy to use with a trained network by just changing the size of the input image. Second, to recognize actions by frame, the change of the frame window size is more flexible (in our case 30).

[Comment 3]

Section 2.3. Have you used dropout for reducing overfitting? Could you specify the tools used for developing the CNN? Are you using keras?

Response) We used dropout with a 70% probability during training and we used a Tensorflow framework. The revised manuscript in Section 2.3 Line 178.

To prevent overfitting, we also conducted dropout after the first, third, and fully connected layers with a 70% probability.

[Comment 4]

Section 3.2. must start with an introductory paragraph.

Response) Thank you for your comment. We revised the section organization as follows.

  1. Experimental setup

3.1 Dataset

3.2 Training

  1. Results and Discussion

[Comment 5]

Section 3.2.1. Have you use the same CNN in all cases? The comparison in this section is focused on the camera? The CNN is the same. Have you tried with other CNNs? I think it is important to analyze the time consuming of the different strategies in order to support your contribution.

Response) Thank you for your constructive comment. We used the same CNN in all cases because other state-of-the-art CNN models had latency during inference on the embedded board. We have added the FPS results for other CNN models in Table 2. Section 4.1 Line 252.

Table 2. Frame per second between models and module power modes.

Model

15W power

FPS

MAX power

FPS

VGG 19 [31]

x

x

Inception V4 [22]

Resnet-50 [23]

Ours

4~5

5~6

7~8

8~9

9~10

14~15

The VGG-19 network [31] has a fully connected layer with a size of 4096, which consumes a large amount of computational memory, making it hard to classify the actions on the Xavier board. Other CNN networks have a skip connection for fast information flow; however, they still have a greater latency than that of our proposed network.

[Comment 6]

You must include results obtained in previous works using NTU-RGBD dataset.

Response) Thank you for your comment. Through the literature survey, the state-of-the-art results of NTU-RGBD dataset show 91.51% [*]. They focused on the accuracy rate on the given data, but we applied the data to real-time human action recognition using the suggested algorithm and mobile robot platform. Their models are also predicting the action labels on full video clips, not as a frame by frame prediction. Therefore, the target application is different and we excluded the literature survey results.

* An End-to-End Spatio-Temporal Attention Model for Human Action Recognition from Skeleton Data- Proceedings of the 31th AAAI Conference on Artificial Intelligence

[Comment 7]

Figure 7 and 8 sizes must be increased. 0% can be removed for a better understanding.

Response)

Thank you for your comment. We revised the figures as follows.

Figure 7. Confusion matrix of Kinect-17.

Figure 8. Confusion matrix of RGB camera-17

Reviewer 3 Report

This manuscript describes a method for human action classification based on a single RGB camera and Convolutional Neural Networks.

The following points summarize my observations about the manuscript and possible revisions:

  1. In the introduction, the following sentence is not clear: “these type of sensors have an on-off functionality, which makes it difficult to recognize detailed actions”. The authors should better define which sensors have an on-off functionality.
  2. In the introduction, the meaning of the following sentence should be explained: “the spatial and temporal information was considered by focusing on frames and joints, respectively”. It Is no clear which the relation between frames and spatial information, and between joints and temporal information.
  3. In the introduction, the following sentence is vague and not supported by evidence. “Besides that, the above studies could not be used for real-time classification, in particular, on the embedded system, because of the complexity of the model structure that is required to achieve a high recognition score for given benchmark datasets”. The authors should be more precise about which methods can or can not have a real-time implementation. It is well known that that the training of a CNN can be very computationally expansive, but a trained CNN can be very fast.
  4. In the introduction, the state of the art (i.e. the related works) is not completely defined. The authors should mention and briefly introduce more known methods, based on visual systems (without a depth sensor) and CNNs, for human gesture or action or posture recognition. With reference to a more thorough state of the art, the authors should highlight the new contributions of the proposed manuscript.
  5.  The first paragraph of section 2.1 repeats some concepts already discussed in the introduction. The authors should move this paragraph to the introduction.
  6. In section 2.1, the contribution of the authors is not clear. The described procedure seems the application of known libraries. The use of a single RGB camera seems a major contribution of the proposed manuscript; thus, this section deserves a better description.
  7. In section 2.2, the authors say that the S matrix is M x N, but we have three columns for each joint. Thus, the number of columns should be N x 3.
  8. In section 2.2, figure 3 and its description are not completely clear and more explanations are needed. For instance, the vector 1 x N is a row vector; thus the stacked M x N image should have M rows (one for each frame, i.e. time instant) and N columns (one for each joint). Figure 3 suggests that the stacked RGB image is a N x M matrix and not vice versa. Moreover, at the end of section 2.1 the authors say that the number of joints is 17, which should be the number of columns (or rows?) of each RGB image depicted in figure 3. However, the number of columns (or rows?) of the RGB images in the bottom part of figure 3 seems higher.
  9. In section 2.2, page 4, line 136, the authors say that one frame (i.e. a single time instant) yields a single row 1 x N x 3 (or column?) of the RGB image, while an action comprising several frames (i.e. several time instants) can be represented stacking the corresponding rows (or columns?) in order to obtain a stacked M x N matrix (or N x M?). In this way, a single stacked matrix is the representation of an action (i.e. a time sub-sequence). The authors should describe how each stacked RGB image can be associated with a single acquired image (i.e. a single time instant), as depicted in figure 3.
  10. In section 2.2, the purpose of the sliding window should be explained.
  11. In section 2.3, the output of the CNN is neither defined nor described,
  12. In section 2.4, figure 6 is not described. The purpose of the tracking is not clear.
  13. In section 3, the sentence “As described earlier, we have sorted action labels …” is not true.
  14. In section 3.1, a clear description of the training and evaluation data sets is missing.
  15. The authors should better define the purposes of the proposed method: action classification from a fixed camera? Action classification from a moving (tracking) camera? Position evaluation of a moving person?

Recommendation

The proposed approach is interesting and the presented results are promising. However, the manuscript should be improved following the listed observations. I recommend not to accept the manuscript in the present version.

Author Response

Reply to the Reviewers' comments

The authors are grateful for the reviewers’ constructive comments regarding the first manuscript. We have done our best to answer all the questions raised by the reviewers, and have revised the manuscript per their suggestions. We have quoted below all the reviewers’ comments in order, and have included our responses (red) and revised manuscript in the paper (blue) after each comment.

Comments of Reviewer 3

[Comment 1]

In the introduction, the following sentence is not clear: “these type of sensors have an on-off functionality, which makes it difficult to recognize detailed actions”. The authors should better define which sensors have an on-off functionality.

Response) Thank you for your comment. We revised the manuscript in Section 1 Line 37.

The sensors monitor activities by achieved sensor data, such as when as a person uses the items, such as opening doors (switch sensor), sitting down on the couch (pressure sensor).

[Comment 2]

In the introduction, the meaning of the following sentence should be explained: “the spatial and temporal information was considered by focusing on frames and joints, respectively”. It Is no clear which the relation between frames and spatial information, and between joints and temporal information.

Response) Thank you for your constructive comment. We wrote the wrong sentence. The spatial and temporal information was considered by joints and frames, respectively. They argued that certain joints and frames have different degrees of importance during the action. Therefore, they designed spatial attention to allocate different attention to different joints within each frame and temporal attention to allocate different attention to different frames during the actions. We revised the manuscript in Section 1 Line 89.

In other words, the spatial and temporal information was considered by focusing on joints and frames.

[Comment 3]

In the introduction, the following sentence is vague and not supported by evidence. “Besides that, the above studies could not be used for real-time classification, in particular, on the embedded system, because of the complexity of the model structure that is required to achieve a high recognition score for given benchmark datasets”. The authors should be more precise about which methods can or cannot have a real-time implementation. It is well known that that the training of a CNN can be very computationally expansive, but a trained CNN can be very fast.

Response) Thank you for your comment. A trained CNN can be fast; however, the embedded board has fewer computational resources than desktop GPUs. We further compared the FPS with other state-of-the-art CNN networks in Table 2. We added the manuscript in Section 4.1 Line 241.

The VGG-19 network has a fully connected layer size of 4096, which consumes a large amount of computational memory, making it hard to classify the actions on the Xavier board. Other CNN networks have a skip connection for fast information flow; however, they still have a latency greater than that of our proposed network.

[Comment 4]

In the introduction, the state of the art (i.e. the related works) is not completely defined. The authors should mention and briefly introduce more known methods, based on visual systems (without a depth sensor) and CNNs, for human gesture or action or posture recognition. With reference to a more thorough state of the art, the authors should highlight the new contributions of the proposed manuscript.

Response) Thank you for your comment. We added related works based on visual systems including those without depth sensors. We described their limits for application to an embedded board system that we are trying to use in a mobile robot platform. We added the manuscript in Section 1 Line 73.

Correspondingly, based on the idea of an attempt to extend 2D convolutional structures to 3D spatiotemporal structures, 3D convolutional networks were developed (C3D) [13]. They found that they have the best temporal kernel length and that the 3D structures outperform 2D convolutional features on video analysis tasks. They also reported the runtime analysis on a single K40 Tesla GPU processing at 313 FPS. However, according to a paper that characterized several commercial edge devices on different frameworks and well-known convolutional networks, the C3D network is computationally intensive and requires high memory usage, which is not possible with the embedded board system.

[Comment 5]

The first paragraph of section 2.1 repeats some concepts already discussed in the introduction. The authors should move this paragraph to the introduction.

Response) Thank you for your constructive comment. We revised the manuscript in Section 1 Line 82

The Kinect camera uses the time-of-flight method to obtain depth information for characterizing the skeleton joints once the pulsed infrared light (emitted by an IR projector) is reflected back to the object. Using this method, it can easily detect 6 people at once and 25 joints per person. Because of this effective 3D pose estimation technique, many benchmark datasets have been made using this camera and several studies have been performed based on RNNs because of its strength in dealing with sequential data and skeletal data.

[Comment 6]

In section 2.1, the contribution of the authors is not clear. The described procedure seems the application of known libraries. The use of a single RGB camera seems a major contribution of the proposed manuscript; thus, this section deserves a better description.

Response) Thank you for your constructive comment. In this study, we present a method that can recognize the actions in real-time on a mobile platform with an embedded system. Other 3D pose estimation methods using a single RGB camera are not adoptable for the embedded board system. They use High Performance Computing (HPC) to achieve a real-time performance. In our embedded system, we integrated two libraries to accomplish real-time pose estimation using a single RGB camera. We revised the manuscript to support our contribution and added in Section 2.1 Line 117.

Recently, there have been various studies for human pose estimation, which is key for the understanding of human behavior on a robotic system. 2D pose estimation has made significant progress via OpenPose [16]. Today, many researchers are devising methods for estimating 3D postures with a single camera. Some applications have been performed; however, there are few methods for real-time performances. Marc Habermann et al. first showed a real-time full body performance capture system using a single camera [17]. They proposed an innovative two-stage analysis that reconstructs the dense, space-time coherent deforming geometry of people in loose clothing. For the pipelined implementation, they used two Geforce GTX 1080Ti GPUs and achieved around 25 FPS. Dushyant Mehta et al. presented a more improved method for a real-time approach for multi-person 3D motion capture at over 30 FPS using a single RGB camera [18]. They contributed a new CNN network called SelecSLS net to improve the information flow. For real-time performances, they used a single Geforce GTX 1080Ti GPU. All these approaches show great real-time performances; however, the hardware setup is not applicable for the embedded system.

[Comment 7]

In section 2.2, the authors say that the S matrix is M x N, but we have three columns for each joint. Thus, the number of columns should be N x 3.

Response) Thank you for your comment. We revised the description M × N to M × N × 3 in Section 2.2 Line 152.

The shape of the joint from a video clip consisting of one action is an M × N × 3 matrix form of equation (1), where M is the number of frames and N is the number of joints.

[Comment 8]

In section 2.2, figure 3 and its description are not completely clear and more explanations are needed. For instance, the vector 1 x N is a row vector; thus the stacked M x N image should have M rows (one for each frame, i.e. time instant) and N columns (one for each joint). Figure 3 suggests that the stacked RGB image is a N x M matrix and not vice versa. Moreover, at the end of section 2.1 the authors say that the number of joints is 17, which should be the number of columns (or rows?) of each RGB image depicted in figure 3. However, the number of columns (or rows?) of the RGB images in the bottom part of figure 3 seems higher.

Response)

Thank you for your comment. We changed the figure as follows.

Formal figure                                            Revised figure

[Comment 9]

In section 2.2, page 4, line 136, the authors say that one frame (i.e. a single time instant) yields a single row 1 x N x 3 (or column?) of the RGB image, while an action comprising several frames (i.e. several time instants) can be represented stacking the corresponding rows (or columns?) in order to obtain a stacked M x N matrix (or N x M?). In this way, a single stacked matrix is the representation of an action (i.e. a time sub-sequence). The authors should describe how each stacked RGB image can be associated with a single acquired image (i.e. a single time instant), as depicted in figure 3.

Response)

Thank you for your comment. We added the following figure for additional detail.

Figure 3. Joint data conversion to RGB image.

[Comment 10]

In section 2.2, the purpose of the sliding window should be explained.

Response) Thank you for your comment. The purpose of the sliding window is to get a predicted action label per frame and further to get more training data. We added the introduction of the sliding window in Section 2.2 Line 161.

In this study, we used a sliding window to generate a subsequence in a video clip that contains one action for data augmentation. We set the stride to 1 for the sliding window to get a predicted action label on each frame, and the window size to 30 frames to execute approximately one to two seconds of movement in an image. Consequently, the action label on the current frame is the result of the previous 30 frames.

[Comment 11]

In section 2.3, the output of the CNN is neither defined nor described,

Response) Thank you for your comment. We added the description in Section 2.3 Line 178.

In the final layer, k action labels were classified by a fully connected layer.

[Comment 12]

In section 2.4, figure 6 is not described. The purpose of the tracking is not clear.

Response) Thank you for your comment. We removed the figure and added the procedure of the tracking algorithm. We revised the manuscript in Section 2.4 Line 193.

These values were estimated from the embedded board and communicated to OpenCR on a serial circuit. The motors were controlled using the tracking algorithm to keep the detected person in the center of the camera screen and maintain the distance between the person and the robot.

[Comment 13]

In section 3, the sentence “As described earlier, we have sorted action labels …” is not true.

Response) Thank you for your comment. We revised Section 3.

3.1. Dataset

The proposed method was evaluated on the NTU-RGBD dataset [28], which is the largest dataset collected by the Kinect V2 camera. It contains approximately 56,000 action sequences from 60 action classes. The most challenging part of this dataset is that it is typically recorded on a variant view that mainly covers three different views (-45, 0, 45). The dataset has four modalities, depth maps, RGB frames, and IR sequences. In our experiments, we have used the RGB frame modality to extract 3D joints by pipeline and trained them on the CNN network. To generate an image, joints from the OpenPose needed to be fully observed, thus we sorted the action classes so that no self-occlusion occurs. The action classes were sorted into 15 labels, such as eating a meal/snack, brushing teeth, drop, sitting down, standing up, hopping, jumping up, making a phone call, playing with the phone, checking time, rubbing two hands together, saluting, putting the palms together, crossing hands in front, and touching chest. As a result, we used a total of 2,217 clips that were further converted to a total of 95,266 images by the sliding window method and we conducted 5-fold cross validation.

3.2. Training

To validate our method, we trained our network for 500 epochs using a stochastic gradient descent [29] optimizer by setting the learning rate to 0.001 and the batch size to 64. To avoid overfitting, we have used L2 regularization [30] with a weight decay of 0.0025. To validate the use of a single RGB camera in the action recognition problem through the learning, we compared two types of joint data that were extracted by the proposed method and by using a Kinect camera. As described earlier, we sorted action labels in which identical self-occlusion did not occur. Moreover, using the Kinect camera output of 25 joints, we set the number of joints to 17 just as with the proposed method. All training processes were run on NVIDIA RTX 2080Ti (11 GB), AMD Ryzen 7 2700, RAM 24 GB in WINDOW OS with a Tensorflow framework.

[Comment 14]

In section 3.1, a clear description of the training and evaluation data sets is missing.

Response) Thank you for your comment. We added a description of the training and evaluation data set in Section 3.1 Line 210.

As a result, we used a total of 2,217 clips that were further converted to a total of 95,266 images by the sliding window method and we conducted 5-fold cross validation.

[Comment 15]

The authors should better define the purposes of the proposed method: action classification from a fixed camera? Action classification from a moving (tracking) camera? Position evaluation of a moving person?

Response) Thank you for your comment. As we described in the Introduction, in this study, we proposed an action recognition system using a single RGB camera for the mobile robot platform with an embedded board. For real-time recognition, we integrated two libraries for 3D joint extraction using a single RGB camera. For the purpose of monitoring a person in an indoor environment, we used a mobile robot platform and tracked a person for continuous detection.

Reviewer 4 Report

In this paper, a new approach for real-time human action recognition using a mobile robotic platform equipped with a monocular RGB camera is proposed. The present method proposes pose-based action recognition, extracting the 3D pose with the use of SoA deep learning methods for 2D pose estimation from RGB (OpenPose) and 3D from 2D pose estimation (3D-pose-baseline). The 3D pose sequences (windows of frames) are mapped on 2D color images and feed a lightweight CNN for action labeling. A person physical tracking is also proposed allowing the robotic platform to follow the person it monitors. The method seems to have practical impact, however, the scientific and research impact is low due to the limited novelty and scientific contribution, since SoA methods are used without extensions and extra contributions. Nevertheless, it is significant that the method presents high action recognition accuracy in comparison with SoA.

Comments:

  • More SoA references to the various research topics involved in this approach could be described:
    • 3D pose estimation from RGB (e.g. XNect: Real-time Multi-person 3D Human Pose Estimation with a Single RGB Camera; 3D Human Pose Estimation with 2D Marginal Heatmaps; LiveCap: Real-time Human Performance Capture from Monocular Video, etc.)
    • Real-time action recognition (e.g. you can take on a look on the survey:  A Comprehensive Survey of Vision-Based Human Action Recognition Methods)
  • The method uses a RGB camera to capture the subject. Then, applying 3D pose estimation from RGB, the pose per frame input is given to the proposed CNN to predict the action label. Since, as presented in comparison with Kinect skeleton tracking, pose accuracy affects the action recognition outcome, comparison between pose estimation SoA methods that could be used on the robotic platform could be examined to showcase that the best pose estimation has been used (i.e. to explain the selection of the pose estimation method).
  • The pose data to image conversion is a technique used already in the literature (e.g. Emotion Recognition from Skeletal Movements). References to such approaches could be added.
  • The tracking algorithm (Figure 6) could be removed.
  • Can you further explain the use of L2 regularization in Line 170? You mean you use L2 loss during supervision?
  • It seems that the results of the method on NTU-RGB-D dataset are favorably comparable against action recognition SoA methods. More results should be included (or even conducted) in order to report the comparison against SoA. 

Other comments:

  • "K" in Figure 4, is the number of action classes? Please give explanation of the variables where needed.
  • All figures are of low quality (stretched, low-resolution, etc.). It is recommended to be replaced with improved figures.

Author Response

Reply to the Reviewers' comments

The authors are grateful for the reviewers’ constructive comments regarding the first manuscript. We have done our best to answer all the questions raised by the reviewers, and have revised the manuscript per their suggestions. We have quoted below all the reviewers’ comments in order, and have included our responses (red) and revised manuscript in the paper (blue) after each comment.

Comments of Reviewer 4

[Comment 1]

More SoA references to the various research topics involved in this approach could be described:

3D pose estimation from RGB (e.g. XNect: Real-time Multi-person 3D Human Pose Estimation with a Single RGB Camera; 3D Human Pose Estimation with 2D Marginal Heatmaps; LiveCap: Real-time Human Performance Capture from Monocular Video, etc.)

Real-time action recognition (e.g. you can take on a look on the survey: A Comprehensive Survey of Vision-Based Human Action Recognition Methods)

Response) Thank you for your comment. We added more references about related topics in Section 2.1 Line 117.

Recently, there have been various studies for human pose estimation, which is key for the understanding human behavior on a robotic system. 2D pose estimation has made significant progress via OpenPose [16]. Today, many researchers are devising methods for estimating 3D postures with a single camera. Several applications have been performed; however, there are few methods for real-time performances. Marc Habermann et al. first showed a real-time full body performance capture system using a single camera [17]. They proposed an innovative two-stage analysis that reconstructs dense, space-time coherent deforming geometry of people in loose clothing. For the pipelined implementation, they used two Geforce GTX 1080Ti GPUs and achieved around 25 FPS. Dushyant Mehta et al. presented a more improved method with a real-time approach for multi-person 3D motion capture at over 30 FPS using a single RGB camera [18]. They contributed a new CNN network called SelecSLS net to improve the information flow. For the real-time performance, they used a single Geforce GTX 1080Ti GPU. All these approaches show great real-time performances; however, the hardware setup is not applicable for the embedded system.

[Comment 2]

The method uses a RGB camera to capture the subject. Then, applying 3D pose estimation from RGB, the pose per frame input is given to the proposed CNN to predict the action label. Since, as presented in comparison with Kinect skeleton tracking, pose accuracy affects the action recognition outcome, comparison between pose estimation SoA methods that could be used on the robotic platform could be examined to showcase that the best pose estimation has been used (i.e. to explain the selection of the pose estimation method).

Response) Thank you for your constructive comment. We have chosen the pose estimation method that can run on the embedded board in real-time. Currently, OpenPose is the state-of-the-art 2D pose estimation that can run on the embedded board. The 3D-baseline also established a strong performance, which affects the action recognition accuracy.

[Comment 3]

The pose data to image conversion is a technique used already in the literature (e.g. Emotion Recognition from Skeletal Movements). References to such approaches could be added.

Response) Thank you for your comment. We added related references in Section 2.2 Line 149.

For this reason, many studies are converting pose sequence data to images to use CNN [24].

[Comment 4]

The tracking algorithm (Figure 6) could be removed.

Response) Thank you for your comment. We removed the figure and added the procedure of the tracking algorithm. We revised manuscript in Section 2.4 Line 193.

These values were estimated from the embedded board and communicated to OpenCR on a serial circuit. The motors were controlled using the tracking algorithm to keep the detected person in the center of the camera screen and maintain the distance between the person and the robot.

[Comment 5]

Can you further explain the use of L2 regularization in Line 170? You mean you use L2 loss during supervision?

Response) Thank you for your comment. We used L2 regularization during the training procedure to avoid overfitting. We revised the Section 3.2 organization to further convey that the l2 loss is used during training.

[Comment 6]

It seems that the results of the method on NTU-RGB-D dataset are favorably comparable against action recognition SoA methods. More results should be included (or even conducted) in order to report the comparison against SoA.

Response) Thank you for your comment. Through the literature survey, the state-of-the-arts results of NTU-RGBD dataset show 91.51% [*]. They focused on the accuracy rate of given data, but we applied the data to real-time human action recognition using the suggested algorithm and mobile robot platform. Their models also predict the action labels on full video clips, not as a frame by frame prediction. Therefore, the target application is different and we excluded the literature survey results.

* An End-to-End Spatio-Temporal Attention Model for Human Action Recognition from Skeleton Data- Proceedings of the 31th AAAI Conference on Artificial Intelligence

[Comment 7]

"K" in Figure 4, is the number of action classes? Please give explanation of the variables where needed.

Response) Thank you for your comment. K is the number of action classes to classify. We revised the manuscript in Section 2.3 Line 178.

In the final layer, k action labels were classified by a fully connected layer.

[Comment 8]

All figures are of low quality (stretched, low-resolution, etc.). It is recommended to be replaced with improved figures.

Response) Thank you for your comment. We improved the figures for better quality.

Round 2

Reviewer 2 Report

I think the paper has improved significantly and I appreciate the effort to improve the presentation and explanations. But my opinion about the impact of the paper is the same. I think the contribution is not very high.

Some additional comments:

  • Table 1 is a mixture of tables??
  • In conclusion section, I’d suggest commenting the main limitations of the proposed system.
  • Section 4. must start with an introductory paragraph.
  • If possible, increase resolution in figures 2 and 4
  • Could you include a photo of the robot ?

Author Response

Responses to the Reviewers' comments

The authors are grateful for the reviewers’ constructive comments regarding the first manuscript. We have done our best to answer all the questions raised by the reviewers, and have revised the manuscript per their suggestions. We have quoted below all the reviewers’ comments in order, and have included our responses (red) and revised manuscript in the paper (blue) after each comment.

Comments of Reviewer 2

[Comment 1]

Table 1 is a mixture of tables??

Response) Thank you for your comment. We have revised the table as an image.

Table 1. Accuracy by camera and number of joints.

[Comment 2]

In conclusion section, I’d suggest commenting the main limitations of the proposed system.

Response) Thank you for your comment. We have revised the manuscript in Line 288.

In addition, the main limitation of this method that recognizing the front view of the person, a more precise pose estimation algorithm with robustness has to be developed for self-occlusion problems.

[Comment 3]

Section 4. must start with an introductory paragraph.

Response) Thank you for your comment. We have added an introduction in Line 226.

In this section, we present the train result of proposed model on NTU-RGBD dataset with 3 types of data RGB camera-17, Kinect-17, and Kinect-25. We also show the tracking results and full system that recognizing the actions while tracking a person with mobile robot.

[Comment 4]

If possible, increase resolution in figures 2 and 4

Response) Thank you for your comment. The figure 2 was already recorded on low resolution. We have increased the resolution of Figure 4.

Figure 4. Converted image of action sequence.

Figure 4. Converted image of action sequence.

[Comment 5]

Could you include a photo of the robot ?

Response) We have revised the figure of the robot as follow

Figure 6. Mobile robot.

Reviewer 3 Report

The modifications introduced by the authors improve the manuscript. However, the following observations should be considered before acceptance:

  1. In section 4.1, table 1 and 2 are not visible.
  2. In the introduction, the original contribution of the proposed manuscript remains unclear.

I recommend to accept the manuscript requiring minor revisions.

Author Response

Responses to the Reviewers' comments

The authors are grateful for the reviewers’ constructive comments regarding the first manuscript. We have done our best to answer all the questions raised by the reviewers, and have revised the manuscript per their suggestions. We have quoted below all the reviewers’ comments in order, and have included our responses (red) and revised manuscript in the paper (blue) after each comment.

Comments of Reviewer 3

[Comment 1]

1.In section 4.1, table 1 and 2 are not visible.

Response) Thank you for your comment. We have revised the table as an image.

Table 1. Accuracy by camera and number of joints.

Table 2. Frames per second between models and module power modes.

[Comment 2]

2.In the introduction, the original contribution of the proposed manuscript remains unclear.

Response) Thank you for your comment. We have revised the manuscript in Introduction section in Line 97.

In this study, we proposed a method that can recognize the human actions in real-time with a mobile robot on embedded system. Considering the size of the mobile robot, we used a single RGB camera instead of the Kinect camera to achieve 3D skeleton joints by integrating the two open source libraries. In this way, the range of use of training dataset increases by using not only the public Kinect dataset but also the dataset that achieved by RGB camera like previously recorded video. To use a simple image classifier of the CNN model for embedded system, we converted the skeleton joint data to image form. The whole process is depicted in Figure 1. Skeletal joints were extracted from the RGB camera image, and their movements were classified using CNN within the embedded board. For continuous observation during indoor conditions, we can track the person using the skeleton joints.

Reviewer 4 Report

The proposed method is novel with respect to the applicability of real-time machine (deep) learning techniques to robotic systems. In my honest opinion, since the method has very limited scientific contribution, the impact of such a method should be strong to the evaluation and presentation section in order to let the readers know the value of this approach. The authors should give a very extended presentation of the results in comparison with the relevant literature, explaining the pros and cons of the method. The evaluation section should be enriched giving evidence that this method, although weak in comparison with the SoA (e.g. comparison of accuracy but also the performance (processing time) on NVIDIA JETSON XAVIER board). To this end, it would be clear for the readers the significance and impact of the proposed method.

Author Response

Responses to the Reviewers' comments

The authors are grateful for the reviewers’ constructive comments regarding the first manuscript. We have done our best to answer all the questions raised by the reviewers, and have revised the manuscript per their suggestions. We have quoted below all the reviewers’ comments in order, and have included our responses (red) and revised manuscript in the paper (blue) after each comment.

Comments of Reviewer 4

The proposed method is novel with respect to the applicability of real-time machine (deep) learning techniques to robotic systems. In my honest opinion, since the method has very limited scientific contribution, the impact of such a method should be strong to the evaluation and presentation section in order to let the readers know the value of this approach. The authors should give a very extended presentation of the results in comparison with the relevant literature, explaining the pros and cons of the method. The evaluation section should be enriched giving evidence that this method, although weak in comparison with the SoA (e.g. comparison of accuracy but also the performance (processing time) on NVIDIA JETSON XAVIER board). To this end, it would be clear for the readers the significance and impact of the proposed method.

Response) Thank you for your comment. Our contribution is to find ways that can apply in real-world using mobile robot with embedded system. As we mentioned on the introduction section, with the development of the sensors that are easy to use in smartphones such as accelerometers and gyro sensors, non-vision based action recognition methods are already adopted in reality. However, indicatable action labels are limited by the types of the sensors. For example, activity recognition methods with accelerometer sensors can only define the actions like sitting, walking, and running etc. In this reason, vision-based approaches emerged and lots of deep-learning networks have been developed. To understand the actions, conventional 2D CNNs have a limitation by the absence of temporal information. Recently, 3D CNNs have been developed that can jointly learn spatial and temporal features but the computation cost is too high, making the deployment on edge devices difficult. Table 2 shows the results. In line 254, the conventional 2D CNNs have a latency on embedded board systems. Other researches called real-time action recognition are experimented on high performance computing (HPC) materials not on the embedded board especially on Jetson Xavier board. Evaluation with other methods are not reachable since they are not using Xavier board. We included other recognition methods on HPC in the introduction section with pros and cons, and mostly cons are presented about processing time. We also introduced other methods that estimating 3D skeleton joints on a single RGB camera which are not adoptable on Xavier board.

Round 3

Reviewer 2 Report

I think the authos have addressed all my comments.

Reviewer 4 Report

Since the proposed method is novel with respect to the practical applicability of real-time machine (deep) learning techniques to robotic systems, it can be considered appropriate for publication in MDPI Sensors.